# Type D Personality and Health Behaviors in People Living with Obesity

**DOI:** 10.3390/ijerph192214650

**Published:** 2022-11-08

**Authors:** Marta Buczkowska, Michał Górski, Joanna Domagalska, Krzysztof Buczkowski, Przemysław Nowak

**Affiliations:** 1Department of Toxicology and Health Protection, Faculty of Health Sciences in Bytom, Medical University of Silesia, 41-902 Katowice, Poland; 2Doctoral School of the Medical University of Silesia in Katowice, Faculty of Health Sciences in Bytom, Medical University of Silesia, 40-055 Katowice, Poland; 3Department of Environmental Health, Faculty of Health Sciences in Bytom, Medical University of Silesia, 41-902 Katowice, Poland; 4Department of General and Oncological Surgery, City Hospital, 41-100 Siemianowice Slaskie, Poland; 5Department of Pharmacology, Faculty of Medicine, University of Opole, 45-052 Opole, Poland

**Keywords:** health behaviors, obese patients, obesity, type D personality, eating behaviors, health control

## Abstract

Background: Considering that health behaviors and personality traits play an important role in the formation of health attitudes, the main objective of this study was to evaluate the relations that occur between type D personality and health behaviors in a group of obese patients. Methods: 443 adult patients with BMI ≥ 30 kg/m^2^, who had been hospitalized in selected hospital facilities in the Silesian Voivodeship (Poland), participated in the study. Respondents completed three standardized questionnaires—the Multidimensional Health Locus of Control Scale, version A (MHLC-A), the Inventory of Health Behaviors (IZZ), and the Type D Scale (DS-14). Results: Patients with type D personality were characterized by the least effective mental attitudes and preventive behaviors, and differed significantly from the other personality types (intermediate and non-type D). Type D personality increased the risk of initiating improper health behaviors by more than five times. Regarding the sense of health control, patients with type D personality had significantly lower scores for the Internal Dimension subscale (21.3 ± 3.1) and higher for the Powerful Others Dimension subscale (24.0 ± 2.6), compared to patients with intermediate and non-type D personality. Proper health behaviors correlated with an internal sense of health control; the strongest correlation, defined as a medium, was with Preventive Behaviors (R = 0.42; *p* < 0.0001). Conclusions: Type D personality was associated with poorer attitudes towards health. Among obese respondents with a type D personality, there was a significantly higher prevalence of those who believed that their health status was a consequence of chance events.

## 1. Introduction

Obesity is a chronic metabolic disease that is the consequence of a sustained positive energy balance over a long period time, leading to excessive fat accumulation [1]. The disease has a measurable impact on physical and mental health and quality of life and generates significant direct and indirect costs to the health care system [2,3,4]. Today, the global number of deaths related to obesity and overweight is higher than those caused by malnutrition (except in sub-Saharan Africa and Asia) [5]. On average, OECD (The Organisation for Economic Co-operation and Development) countries [6] spend 8.4% of their health budget to provide treatment for obesity-related diseases and their consequences, corresponding to approximately USD PPP (purchasing power parity) 311 billion per year (or USD PPP 209 per capita per year). The most important consequences of obesity are chronic diseases: diabetes, cardiovascular disease, and cancer [7]. By 2050, obesity will be responsible for 70% of all diabetes treatment costs, 23% of cardiovascular disease treatment costs, and 9% of cancer treatment costs [6]. Obesity and related diseases will reduce life expectancy by about three years in OECD, EU-28 and G20 countries from 2020 to 2050 [6]. In Poland, 20% of men and 18% of women are obese, and 46% and 31% overweight, respectively, or an average of 57% of Polish adults have above-normal body weight [8]. Unfortunately, the only way to reduce these costs is through appropriate obesity prevention, in which the mental aspect of the disease may be particularly important [9].

Research conducted for several decades indicates that the type of personality may be important in the progression of many diseases. Currently, it has not been possible to link obesity to a specific personality type, but some characteristics of obese people have been observed, including passivity, indecisiveness, pessimism, or difficulty demonstrating emotions [9]. Obese people are also negative about their appearance and, consequently, are more likely to experience stressful situations due to that fact. This decreases their self-esteem, which is also negatively affected by the patient’s surroundings that do not support or regard overweight people positively through the prism of social stereotypes [10]. Obese people are very often perceived as lazy, not self-caring, and unambitious, which makes them less likely to be hired, despite possessing the same skills as people of normal weight. Unsuccessful attempts to change eating habits or reduce body weight further project a lower sense of self-efficacy, perpetuating the belief that they do not have influence over [9,11,12] the situation [11]. Studies (based on the five-factor personality model) have shown that higher conscientiousness is associated with lower BMI values, a lower risk of obesity, and less weight gain in adulthood [13,14]. Moreover, conscientious individuals show better health behaviors, with regard to preventive activities, and are more likely to engage in physical activity [15]. The relationship between neuroticism and body weight is apparently curvilinear; a greater severity of neurotic traits is found with both too low and too high BMI. Thus, neuroticism can be associated with both underweight and overweight [16,17]. Longitudinally, low agreeableness and impulsivity-related traits predicted a greater increase in BMI in adulthood [18]. Dependencies between personality dimensions and eating habits were also described [19]. Perhaps the most consistent predictor of healthy eating habits is openness; people who scored higher for this personality dimension were more likely to choose and consume so-called “healthy foods” [20]. Also suggested is a small positive relationship between conscientiousness and healthy eating; more conscientious people are reluctant to take risks, including “dietary risks” [11]. Many traits observed in the obese population may indicate a higher prevalence of type D personality in this group. Type D personality may have a greater predictive value for obesity compared to selected traits because, as a personality construct, it tends to be more stable over time [21].

The type D personality, or distressed personality, is the latest distinguished type of personality. The term was defined in 1995 by the Dutch clinical psychologist Johan Denolett [22]. Type D is characterized by a tendency to hold back from expressing personal feelings (social inhibition) with simultaneous strong experiencing of negative emotions (negative affectivity). Distressed personality shares with neuroticism a susceptibility to stress, a tendency to break down in stressful situations and to experience severe anxiety, and pessimism in the assessment of reality, whereas social inhibition has many traits in common with introversion, including relationship difficulties or the tendency to be alone [23]. This results in patients with this personality type being less able to cope with the symptoms of the disease and its consequences [22,24]. Potential mechanisms through which the type D personality [25,26] may have negatively impacted health include:health behaviors—studies showed that people with type D personality were less likely to exercise, follow a healthy diet, take medications as recommended, and regularly attend medical check-ups, compared to their counterparts with a non-type D personality [27,28,29];increased emotional stress—type D personality has been associated with depression, anxiety, and other indicators of emotional stress in individuals with diseases and in the general population [24,30];standard risk factors—some studies observed a link between type D personality and standard risk factors such as blood pressure, cholesterol levels, and above-normal body weight [31,32], while others did not [33].

Type D can be a risk factor for the development of many diseases, including cardiovascular disease [34,35,36], psoriasis [37], and some cancers [38,39,40]. In 2012, type D personality was included in the European Association of Preventive Cardiology (EAPC) guidelines as an established risk factor for coronary heart disease (CHD [38,41,42,43]) [34]. Regarding obesity, it was observed that in women, both type D personality dimensions correlated with a higher percentage of visceral fat [44]. Furthermore, it was found that negative affectivity was an independent risk factor in the pathogenesis of obesity, hypertension, and reduced HDL cholesterol levels [45,46]. However, the number of studies devoted to type D personality in the obese population is limited.

The mental aspect of obesity is complicated, as mental disorders cannot only be a cause of the disease but can also develop or be exacerbated as a consequence. Cross-sectional studies indicate that people who were experiencing stressful life events had high levels of perceived stress characterized by higher BMI than those without such experiences [47,48,49]. On the other hand, excessive caloric intake, usually one of the main causes of obesity, negatively affects the functioning of the whole body, including the brain, inducing neurodegenerative changes [50,51]. The underlying [51] neurodegenerative damage associated with obesity involves inflammatory processes, oxidative stress, and mitochondrial dysfunction, which are exacerbated [52]. Studies in animals with diet-induced obesity showed significant changes in the hippocampus and frontal cortex, including microglial activation, the dysfunction of transient receptor potential (TRP) ion channels, or lower expression of choline acetyltransferase (ChAT) and the vesicular acetylcholine transporter (VAChT) [53,54]. Thus, it would appear that obesity, by the mechanisms described, might also affect cognitive dysfunction [54].

Health behavior is directly related to obesity [55]. The concept of health behavior is complex. Gochman [56] and Parkerson [57] provided key contributions to defining the concept of health behavior. According to Gochman [56], health behaviors should be understood as personal characteristics such as beliefs, expectations, values, and other cognitive elements, personality traits (including emotional and affective states), and overt patterns of behavior, actions, or habits related to maintaining, restoring, and improving health. Parkerson [57] thought that health behavior should be viewed in a more general sense, as the actions of individuals, groups, and organizations that correlate with and influence social change, the creation and implementation of policy agendas, and the improvement of quality of life. In contrast, one of the basic definitions of health behaviors states that they are actions taken by an individual to maintain or improve health, achieve a positive body image, or prevent health problems [58].

A difficulty in defining health behaviors has influenced differences in the way they are classified. In 1966, Kasl and Cobb [59] were the first to define three categories of health behaviors: preventive health behaviors, illness behaviors, and sick role behaviors. In the 1990s, Schwarzer [60] originally implemented a widely established simplified classification of health behaviors, thus distinguishing between health-promoting and risky behaviors. In 2003, Juczynski [61] proposed a different structure of health behaviors, identifying only groups of health-promoting behaviors, such as health practices, safety practices, preventive practices, avoidance of environmental risks, and avoidance of harmful substances [62]. Some studies concentrate only on healthy lifestyle behaviors, distinguishing them into the following categories: physical activity, eating behavior, alcohol consumption, sleep disturbance, and smoking [55,63].

Health behavior is strongly linked to our lifestyle, which is derived from individual predispositions and social and cultural conditions [55]. Health habits are formed during early life, influencing later behavior and, consequently, the health of the adult [64]. The health behaviors undertaken depend on age, gender, education, marital status, family, and financial situation and occupation [65,66]. Personality traits also play an important role in determining health behavior [67]. It has been observed that risky behaviors are associated with low stress tolerance, emotional immaturity, difficulty expressing feelings, low self-esteem, feelings of loneliness, and high levels of anxiety [68,69,70]. On the other hand, pro-health behaviors are positively correlated with a sense of coherence, optimism in life, self-efficacy, and an internal locus of control over health [58,71].

Given the association of type D personality with the least effective attitudes toward health, it was decided to examine the extent to which this personality construct could be used in obese patients, both to better adjust clinical intervention, and also to prevent the disease itself, or its consequences. The authors decided to study among obese patients because of the limited amount of distressed personality research on this group and the traits and attitudes common among those with above-normal body weight and type D personality. Thus, the main purpose of this study was to evaluate the relationships that exist between type D personality and health behaviors in a group of obese patients in Poland, where more than half of all adults have above-normal body weight. Furthermore, an attempt was made to determine the sense of health control as a determinant of selected health behaviors, taking into account the personality types analyzed.

## 2. Materials and Methods

### 2.1. Study Group

The study group was adult patients with BMI ≥ 30 kg/m^2^, who were hospitalized in selected hospitals in Silesia Voivodeship, Poland.

The survey of the study was conducted between May 2018 and December 2019. A total of 443 correctly completed questionnaires were analyzed. Based on the unknown fraction of type D personality in the population of hospital patients, and assuming an effect size (fraction) of at least 0.1 [72], the minimum sample size was set at 440.

Exclusion criteria were a condition that made it impossible to complete the questionnaires, including severe disease accompanied by fever or a postoperative condition, dependence, and symptoms of impaired cognitive functioning (autopsychic and allopsychic orientation disorders identified by screening).

### 2.2. Ethics Approval

The study design was approved by the directors of the institutions in which the study was conducted.

Bioethics Committee approval was not required, due to the questionnaire type of this study (opinion dated 23 May 2018 No.: KNW/0022/KB/106/18). According to Polish law, this study was not a medical experiment, so it did not require the consent of the Bioethics Committee (Act of 5 December 1996, on the professions of physician and dentist (i.e., Journal of Laws 2019, item 537). Nevertheless, all research standards were observed in the study. It complies with the provisions of the Declaration of Helsinki. Patient contribution to the study was voluntary. The authors used no patient identification or confidential data from medical records. The headline of each survey contained a note to the patient that the survey was anonymous and the results would be used for research purposes.

### 2.3. Research Tools

Patients, with voluntary participation, completed the authors’ survey and standardized questionnaires: The Multidimensional Health Locus of Control Scale, version A (MHLC-A), The Inventory of Health Behaviors (IZZ), and Type D Scale-14 (DS-14).

#### 2.3.1. The Authors’ Survey

The authors’ survey was composed of 8 questions relating to socio-demographic data (gender, age, education, occupational activity status, place of residence, marital status), as well as respondent weight and height.

#### 2.3.2. The Inventory of Health Behaviors (IZZ)

The IZZ by Juczynski [73] was developed to determine the level of health-promoting behaviors. The IZZ questionnaire consists of 24 statements characterizing different types of health-related behaviors. The tool determines overall health behaviors (Overall Health Behavior Index, ZZ) and the level of four dimensions of health behaviors, which include:Proper Eating Habits (PN1)—a dimension related to the type of food consumed;Positive Mental Attitude (PN2)—a dimension that includes psychological factors such as susceptibility to stress;Health Practices (PZ)—a dimension related to daily sleep or physical activity habits;Preventive Behaviors (ZP)—a dimension relating to respect for health recommendations and self-inquiry about health and disease.

Completion of the Inventory is based on determining the frequency of selected health-related activities. This is accomplished using a five-point scale, in which activities that are occasionally undertaken are given a 1 (almost never) and those that are very common a 5 (almost always). The remaining points on the scale correspond to behaviors of moderate frequency: 2—rarely, 3—occasionally, and 4—frequently. Respondents, taking into account the possibility of periodic changes in some healthy habits, when they complete the questionnaire, should consider only the last year. The Overall Health Behavior Index (ZZ), which is the sum of all scores, ranges from 24 to 120 points. A higher score on this index indicates a higher intensity of health-promoting behaviors. The ZZ index is transformed into standardized units that are then assigned specific values on a sten scale (1–10) to better interpret the results obtained. This transformation is based on the results of the normalization group, separate for men and women (Table 1). Scores of 1–4 sten are regarded as low, 5–6 as average, and values of 7–10 are defined as high.

In addition, the severity of health behaviors can be determined in each of the four categories based on the average number of points for items with specific numbers in the Inventory: 1, 5, 9, 13, 17, 21—Proper Eating Habits; 2, 6, 10, 14, 18, 22—Preventive Behaviors; 3, 7, 11, 15, 19, 23—Positive Mental Attitude; 4, 8, 12, 16, 20, 24—Health Practices.

The reliability for the entire Inventory (ZZ index) based on Cronbach’s alpha coefficient is 0.85, while for individual subscales it ranges from 0.60 to 0.65 [73].

#### 2.3.3. The Multidimensional Health Locus of Control Scale, Version A (MHLC-A)

The MHLC-A by Wallston, Wallston, and DeVillis [74], is a Polish adaptation by Juczynski [73]. The MHLC scale has two versions, A (MHLC-A) and B (MHLC-B), which are considered equivalent. This study used version A. The MHLC contains 18 statements about an individual’s expectations in three dimensions of locus of health control:Internal Dimension—control over one’s health depends on the individual;Powerful Others Dimension—self-health is the result of the influence of other people, especially medical personnel (External Dimension);Chance Dimension—the individual’s health is the result of chance and other factors of an external nature (External Dimension).

The respondent who completes the questionnaire defines his attitude to the statements presented using a six-point scale, where: 1—strongly disagree, 2—somewhat disagree, 3—somewhat disagree, 4—somewhat agree, 5—somewhat agree, and 6—strongly agree. The final score is the sum of the points obtained for each subscale: Internal Dimension (sum of scores from questions numbered: 1, 6, 8, 12, 13, 17), Powerful Others Dimension (sum of scores from questions numbered: 3, 5, 7, 10, 14, 18), and Chance Dimension (sum of scores from questions numbered: 2, 4, 9, 11, 15, 16). The higher the score, the stronger the perception that a particular factor affects the health of the individual being studied. The internal location of control is assumed to be more beneficial, as it is conducive to health-promoting activity and more responsibility for one’s health.

The reliability of the MHLC-A scale is determined by the value of Cronbach’s alpha coefficient of: 0.74 (Internal Dimension), 0.69 (Chance Dimension), and 0.54 (Powerful Others Dimension) [73].

#### 2.3.4. Type D Scale-14 (DS-14)

The DS-14 [73,75] by Denolett [75] was used to measure type D personality, in a Polish adaptation by Juczynski [76]. This contains 14 statements, including 7 relating to Negative Emotionality (numbers 2, 4, 5, 7, 9, 12, 13), and 7 relating to Social Inhibition (numbers 1, 3, 6, 8, 10, 11, 14). The respondent assesses each statement according to a five-point scale (0-false, 1-totally false, 2-difficult to say, 3-totally true, 4-true). The scores of statements 1 and 3 must be recoded. The final score is the sum of the scores obtained for both subscales, Social Inhibition and Negative Emotionality. The higher the score, the more severe the traits that comprise a given personality dimension. The interpretation of the results for DS-14 is presented in Table 2. The Cronbach’s alpha coefficient of DS-14 is 0.86 for the Negative Emotionality scale, and 0.84 for Social Inhibition [75,76].

### 2.4. Statistical Analysis

In this study, there are detailed characteristics of the patient group by gender and type of personality. Data are presented as mean (X) with standard deviation (SD) or median (Me) with quartiles (Q1–Q3). The prevalence of responses was described by the number n and also expressed as percentages from the total study group. The normality of the distributions was tested using the Shapiro–Wilk test [77]. Homogeneity of variance was assessed using Levene’s test [78]. Tukey’s correction [79] was used to control for statistical significance in multiple comparisons. In the analysis of correlations between the study variables, Spearman’s nonparametric correlation test [80] was used. The effect of independent variables on the dependent variable was assessed using logistic regression analysis, the results of which were presented as odds ratio values. Results for which *p* < 0.05 were considered statistically significant.

Statistical analysis was performed with STATISTICA 13.0 PL (StatSoft Poland, Krakow, Poland) [81,82].

## 3. Results

### 3.1. Characteristics of the Study Group

The study included 443 participants, 59.8% (*n* = 265) of which were female and 40.2% (*n* = 178) male. The mean age of all study participants was 49.6 ± 17.5 years. The highest percentage were patients in the age ranges: 31–40 (*n* = 88; 19.8%); 41–50 (*n* = 95; 21.5%) and 51–60 (*n* = 93; 21%). However, those in the oldest (>70 years and 61–70 years) and youngest (≤30 years) groups accounted for a lower proportion of those surveyed, *n* = 51; 11.5%, *n* = 62; 14%, and *n* = 54; 12.2%, respectively.

Regarding the place of residence, about half of the group were patients who said they lived in a city with a population of 100,000 to 250,000 residents (*n* = 223; 50.3%), followed by those living in cities with more than 250,000 residents (*n* = 99; 22.4%). Another group was made up of residents of villages, with 14% (*n* = 62), followed by city residents in the range of 50,000 to 100,000 residents (*n* = 36; 8.3%) and those living in a city with a population of less than 50,000 (*n* = 22; 5%).

Marital status was another aspect that was included in the characteristics of the respondents. The largest group was the respondents who were married, which was half of the study participants (50.6%). Respectively, 17.4% (*n* = 78), and 17.7% (*n* = 77) were divorced or single. Widowed individuals accounted for 8.1% (*n* = 36) of the total study group, and 5.8% (*n* = 26) declared that they were cohabiting. Only two people (0.4%) reported that they were separated.

The characteristics of the participants also included their educational level. A total of 44.9% (*n* = 199) of the respondents declared that they had secondary education and 40.4% (*n* = 179) that they had higher education. The fewest number of survey participants declared having vocational education—12.2% (*n* = 54), and primary education— 2.5% (*n* = 11).

The other aspect analyzed was the status of occupational activity. Most people, 58.2% (*n* = 258), were employed, followed by pensioners (*n* = 107; 24.2%), the unemployed—7.7%, and students—2.4%. In the survey, 7.5% (*n* = 33) of the people declared another occupational activity status.

The mean height was 169.2 ± 8.09 cm, with the female group at 164.4 ± 5.66 cm and the male group at 176.4 ± 5.41 cm. The mean body weight for the total study group was 111.89 ± 17.99 kg, including 106.54 ± 15.90 kg among women and 119.85 ± 18.03 kg in the case of men. The average Body Mass Index (BMI) was 39.00 ± 5.25 kg/m^2^; females had a BMI of 39.3 ± 5.12 kg/m^2^, and males 38.48 ± 5.41 kg/m^2^.

### 3.2. Health Behaviors Occurring among Obese Patients Due to Type D Personality

A subsequent analysis focused on the health behaviors of the study participants. For this objective, the Health Behavior Inventory (IZZ) was used [73]. The results obtained were compared with the normative values. The mean values of the Overall Health Behavior Index (ZZ), for the study and normalization groups, were at similar levels without statistical significance, while the individual subcategories of health behavior had statistically significant differences [73] (Table 3).

There was an analysis of the differences that occurred between the ZZ and the type of personality. Significantly higher scores were observed for non-type D personality, followed by intermediate personality, while the lowest scores characterized patients with type D personality (Figure 1).

There was also a study of whether there were significant differences in the IZZ subscales in patients with different personality types.

Regarding Proper Eating Habits, the non-type D personality had a mean score of 3.58 ± 0.6, those with an intermediate personality scored 3.45 ± 0.6, and respondents with a type D personality obtained a score of 3.53 ± 0.5. No statistically significant differences were observed.

For Preventive Behaviors, the non-type D personality had the highest score of 3.3 ± 0.5, followed by a lower score for those with an intermediate type personality of 3.1 ± 0.5, and the lowest score was obtained by individuals with a type D personality, of 2.9 ± 0.4. The results obtained were significantly different from each other, as shown in Figure 2a.

The highest scores for the Positive Mental Attitude subscale were obtained by those with a non-type D personality, 3.8 ± 0.3, a lower score was characteristic of participants with an intermediate personality, 3.3 ± 0.4, and the lowest score was achieved by individuals with a type D personality, 3.0 ± 0.4. The observed differences were statistically significant (Figure 2b).

In the Health Practices subscale, the highest scores were obtained by those with a non-type D personality, of 3.52 ± 0.5, followed by those with a type D personality, of 3.49 ± 0.5, and the lowest with an intermediate personality, of 3.42 ± 0.4. However, these differences were not statistically significant.

Correlations between health behaviors and BMI were also evaluated. The results showed that there was no or low effect of health-promoting behaviors on BMI values: for Proper Eating Habits R = −0.01; *p* = 0.7, for Preventive Behaviors R = −0.14; *p* = −0.1, for Positive Mental Attitude R = −0.1; *p* = 0.02, and for Health Practices R = −0.08; *p* = 0.06.

The relationship between health behaviors and a class of obesity was also analyzed. It was shown that patients in all obesity classes most commonly represented average health-promoting behaviors. For people with a BMI of 30–35 kg/m^2^ it was 70% (*n* = 86), for those with a BMI of 35–40 kg/m^2^ it was 66% (*n* = 76), and people with BMI > 40 kg/m^2^ represented 66% (*n* = 135). However, the highest proportion of patients with the lowest level of health-promotion behaviors was found in the morbid obesity group, 69% (*n* = 40). The differences were statistically significant (*p* = 0.0002).

Multivariable analysis was also performed for the results obtained from the IZZ questionnaire. Age and personality type were found to be significant factors in reducing the frequency of engaging in health-promoting behaviors. Type D personality increased the risk of not engaging in health-promoting behaviors by 5.5 times, while age over 71 years increased this risk by 25 times. (Table 4).

### 3.3. Health Locus of Control among the Obese Due to Type D Personality

Further analysis included MHLC-A [73]. The mean scores for each subscale were as follows: 22.7 ± 3.4 (95% CI: 22.4–23.4) for Internal Dimension; 22.7 ± 3 (95% CI: 22.4–23.0) for Powerful Others Dimension; 22.3 ± 3.6 (95% CI: 21.9–22.6) for Chance Dimension.

The average scores obtained in each MHLC-A subscale were analyzed to gender and selected characteristics of the study group. Gender and age were identified as factors significantly differentiating the results obtained in selected subscales. In the Powerful Others Dimension subscale, women had significantly higher scores (23.2 ± 3.0) than men (22.1 ± 3.9). Meanwhile, scores on the Internal Dimension and Chance Dimension subscales were strongly associated with age; as one aged, the internal locus of health control decreased and Chance Dimension increased (Table 5).

Regarding the Internal Dimension subscale, the non-type D personality had a mean score of 24.4 ± 3.4, the intermediate personality 22.6 ± 3.0, and the type D personality 21.3 ± 3.1 (Figure 3a).

On the Powerful Others Dimension subscale, the average highest score was obtained by the non-type D personality (23.1 ± 2.7), followed by the intermediate personality (22.7 ± 3.4), and the lowest score was characterized by the type D personality (22.3 ± 3.1). These differences proved statistically insignificant.

The highest mean score on the Chance Dimension subscale was obtained by those with type D personality (24.0 ± 2.6), followed by those with intermediate personality (22.4 ± 3.6), and finally by respondents with non-type D personality (20.2 ± 3.6). These differences were statistically significant (Figure 3b).

No significant relationship was found between the class of obesity and the health locus of control.

### 3.4. Evaluation of the Correlation between MHLC-A and IZZ Scores

This study also evaluated the correlation between MHLC-A and IZZ. All health behaviors were assessed in the context of the Internal Dimension, Powerful Others Dimension, and Chance Dimension. A weak but highest correlation was recorded between Internal Dimension and Proper Eating Habits (R = 0.29; *p* < 0.001), no correlation was observed between Proper Eating Habits and Powerful Others Dimension (R = 0.01; *p* = 0.8), while a negative weak correlation was found between Proper Eating Habits and Chance dimension (R = −0.15; *p* = 0.001). Concerning Preventive Behaviors, the strongest correlation, defined as a medium, was observed with Internal Dimension (R = 0.42; *p* < 0.0001), followed by a weak correlation with Powerful Others Dimension (R = 0.1; *p* = 0.003) and a weak negative correlation with Chance Dimension (R = −0.25; *p* < 0.001). The assessment of the Positive Mental Attitude subscale showed the strongest correlation with Internal Dimension (R = 0.48; *p* < 0.0001), a weak correlation with Powerful Others Dimension (R = 0.12; *p* = 0.006), and a medium negative correlation with Chance Dimension (R = −0.37; *p* < 0.0001). In the analysis of Health Practices, the strongest correlation was with Internal Dimension (R = 0.33; *p* < 0.001), there was no correlation with Powerful Others Dimension (R = 0.03; *p* = 0.4), and a weak negative correlation with Chance Dimension (R = −0.1; *p* = 0.02).

## 4. Discussion

Health behaviors, like personality, are formed at different life stages and can directly or indirectly affect health in the short- and long-term perspectives [64]. Engaging in health-promoting behaviors can prevent many chronic diseases, such as cancer, heart disease, stroke, and diabetes, thereby reducing the risk of premature death and improving physical and mental health [7]. Proper health behaviors are also crucial in the prevention, or eventual treatment, of obesity and related diseases [55,63]. In 2013, the Institute of Medicine of the National Academies indicated that population-based health-promoting strategies focused on physical activity, or a healthy diet, can effectively counteract excess body weight. Published in 2020, a meta-analysis of overweight and obesity (28 research articles from 2012 to 2019) showed that the primary identified risk factors for above-normal body weight are smoking, improper eating habits, including excessive caloric intake, the consumption of sugary drinks and fast food, low socioeconomic status, sleep disturbances and physical inactivity [83]. However, the maintenance of normal body weight requires multi-dimensional activities, not just selective implementation of certain behaviors in daily life. This can be observed regarding physical activity. It was found that performing 30 min of moderate exercise five times a week has a more beneficial effect on health than dietary supplements and pharmaceuticals used in the prevention and treatment of chronic diseases. However, researchers highlight that self-implemented physical activity does not play a clear role in obesity prevention [84]. For this reason, tools are being used in the analysis of health behaviors that include their various categories.

In the original study, one of the determinants of health behavior was the Overall Health Behavior Index (ZZ), which was at an average level (81.4 ± 9.0, equivalent to 5.49 sten). Respondents showed the best results for Proper Eating Habits (3.5 sten) and the worst for Preventive Behaviors (3.1 sten). The results from other studies relating to above-normal weight individuals are varied. The ZZ is within 5–6 sten, which corresponds to the average level of attitudes that promote health, similar to our study. Some discrepancies are evident in the mean values of the ZZ index, which achieved a lower (78.57 ± 12.37) [85] or higher (84.53 ± 15.03) [86] value, compared to the present analyses. These differences may be due to the peculiarities of the study group, which, in the case of the lower value of ZZ, consisted only of people with obese class II and III (BMI ≥ 35 kg/m^2^), while the higher value of the ZZ characterized patients with BMI ≥ 25 kg/m^2^. Disparities in the overall health behavior index are reflected in the results achieved by respondents for individual subcategories. Patients with a BMI ≥ 35 kg/m^2^ represented worse eating habits (Proper Eating Habits subscale) (2.77 sten) [85] than those with a BMI ≥ 25 kg/m^2^ (3.45 sten) [86] and respondents from the authors’ study, with a BMI ≥ 30 kg/m^2^ (3.5 stena). Sekuła et al. [85] explain the low results achieved by the study group by a stronger tendency to habitual eating if they are morbidly obese, defined as those with BMI ≥ 40 kg/m^2^ [87], compared to those with lower body weight. The original studies also showed that patients representing the lowest level of health-promoting attitudes are most often those with morbid obesity (69%) [85,86]. However, the level of health behavior varies not only among patients with abnormal body weight but also in the general population. Juczynski [73] reported that the average ZZ for adults is 81.82 ± 14.16, while in a study of randomly selected residents of the Silesian Voivodeship [88], the average index was equal to 78.11 ± 16.45 [73,88]. The discrepancies found in the level of health behaviors in the population can be determined by gender (women tend to score higher than men), age (older people have higher levels of health behaviors), or the health status of respondents (for example, diabetics and women with complicated pregnancies show higher levels of health behaviors), among other factors [73].

There are a variety of psychological factors, including the patient’s personality, that can have a significant impact on the uptake of health-promoting attitudes [67,89]. In the present study, type D personality was found to increase the risk of engaging in improper health behaviors by more than five times. In addition, the results indicate that patients with distressed personality represent the least effective mental attitude (3.0 sten) (Positive Mental Attitude subscale) and the least effective preventive behaviors (2.9 sten) (Preventive Behaviors subscale) and are significantly different in this regard from the other personality types (intermediate and non-type D). In contrast, for eating habits and health practices, the results achieved by those with type D personality (3.53 and 3.49 sten, respectively) are at an average level. Comparing the authors’ results with other studies, it can be seen that type D personality is always associated with poorer health care, but there are some differences in the levels of particular types of behavior.

In 2008, it was first shown that among healthy adults there is a relationship between distressed personality and less frequent adoption of health-enhancing behaviors, such as less frequent regular medical checkups [90]. Then, Gilmour and Williams [68] (questionnaire used: Preventive Health Behaviours Checklist-PHBC) found that healthy individuals with type D personality, more often than those with non-type D (non-type D personality plus intermediate personality), engaged in improper health behaviors, which included smoking, unhealthy diet, and lack of physical activity [68]. A 2015 meta-analysis found that personality influenced physical activity in one in ten people in the population and that neuroticism, a trait of type D personality, negatively correlated with physical activity levels, but this relationship varied for gender, age, or geographic area [91]. Studies conducted with cardiac patients confirmed earlier reports type D was positively associated with a sedentary lifestyle, nicotinism, alcohol consumption, and negatively with so-called healthy eating [69,92,93]. The relationship between type D personality and alcohol consumption has been subjected to a more detailed analysis. Bruce et al. [94] concluded that people with higher scores obtained for both personality dimensions, i.e., negative emotionality and social inhibition, showed higher levels of alcohol addiction. The authors found that alcohol consumption was a way to manage negative emotions [94]. Williams et al. [95] obtained similar results; type D was associated with higher alcohol consumption and higher levels of alcohol desire compared to non-type D personality. However, unlike the study by Bruce et al. [94], the presence of a stressor was not found to significantly determine the desire to consume alcohol; the level of alcohol thirst was higher in type D personality people, regardless of the stressor [95]. A German study reported that patients with diabetes and type D personality did not respect the prescribed healthy diet and avoided contact with health care professionals. They were three to four times more likely to engage in improper health behaviors than those with a non-type D personality, which is in line with the authors’ results [1,69]. Particularly important for determining the relationship between distressed personality and health was a 2016 study. Williams et al. [70] proved that inappropriate health behaviors were mediating the link between distressed personality and a poorer subjective assessment of health, projecting poorer quality of life and more physical symptoms, including sleep problems and headaches. However, the authors were unable to explain the mechanism of the associations that occurred [70]. In 2018, Kwon and Kang [96], based on a study among patients with coronary artery disease, suggested that disease perception underlies the link between distressed personality and unhealthy behavior. Type D personalities have been reported to have lower disease perceptions, including greater anxiety and emotional distress. The relation between perceptions of disease and health behaviors is explained by the health theory of self-regulatory systems; when confronted with an illness or other health risk, our behaviors are adapted to the new situation and then they are subject to self-assessment for effectiveness. This leads to the modification of emotional and cognitive responses, as well as health behaviors themselves, which can have important implications for the treatment process [96].

The variety of activities and attitudes that make up health behavior means that they are determined by many factors. Gender and socioeconomic status have been identified as major determinants of smoking and physical activity, while environmental factors influence diet and alcohol consumption. Age is also an important determinant of many health-promoting activities [66,97]. The present study showed that the level of health-promoting behavior decreased with age. The most effective health behaviors were represented by those younger than 36, and the least effective by patients older than 71, for whom the risk of engaging in unfavorable behaviors increased 25 times. [98,99]. Similar results were presented by Ek [99], finding that younger adults (18–35 years of age) are more likely to engage in risky health behaviors, while older adults (51–65 years of age) show high health consciousness, and consequently they have better health behavior patterns [99]. In terms of physical effort, it was observed that the recommended levels of physical activity were most rarely achieved in the group over 65 years old and then the group 45–64 years old. The most active were those in the 18–44 age group [100]. The discrepancies in the results obtained for the older population, in the original study, and from other research, may be due to the very high prevalence of distressed personality in the over 61 age group, which was at 47% (*n* = 54). Type D personalities perceive their surroundings as not providing social support, and as a result, are reserved in their social interactions. In the case of the elderly, this is of particular importance, as they are more likely to be lonely due to their age, which, combined with their personality traits, can lead to complete social isolation. Researchers have shown that both loneliness and social isolation are independent predictors of poor health and mortality, even after taking into account prominent behavioral traits or biological factors [101]. Social isolation also directly affects health behaviors. A meta-analysis published in 2017 suggested that older people with more social support are significantly more likely to engage in physical activity [102]. It also found that singles might be more likely to initiate harmful behaviors, such as smoking, excessive alcohol consumption, and overeating, as a mechanism of psychological relief [103].

The health locus of control can be an important determinant of health behavior. The concept refers to an individual’s belief that one’ s health is controlled by one’s own behaviors (internal locus) or is a consequence of chance events or the influence of others (external locus) [104]. Scientific data indicate that the internal health locus of control is directly associated with better health, both physically and mentally, and with a more frequent presentation of health-promoting attitudes. Individuals with a low Internal health locus of control had significantly higher mortality [105]. The conviction that chance is important in forming health is correlated with poorer health, including a higher prevalence of mental disorders and a more widespread presentation of improper health behaviors, whereas external localization of health control, in which responsibility for one’s health is attributed to others, is reflected in high levels of physician adherence, but may positively correspond with the risk of chronic pain and/or disability. It is thought that preventing risk behaviors is more effective when the individual is convinced of one’s impact on health [106,107].

The health locus of control was also addressed in the present study. The average scores obtained by respondents were similar for all subscales, but a detailed analysis revealed the presence of some differences. It was observed that the means for Internal Dimension were highest for those under 40, while respondents over 61 had higher scores for the Chance Dimension. The results of the Powerful Others Dimension showed significant gender differences; the average scores achieved by women were lower than those achieved by men. While studies regularly confirm the relationship between age and the health locus of control, there is some variation. According to some authors, the external localization of health control increases with age, but without significant changes in internal localization [108,109,110]. There are also studies whose results are consistent with the present one; the internal health locus of control consistently decreases with age, with a concomitant increase in external control [73,111,112]. Changes in the health locus of control are thought to be age-related, due to the characteristics of the aging process; late adulthood is associated with more medical conditions, impaired physical ability, and consequently, poorer health. The individual’s confidence in their own abilities and feelings of control over a situation decrease with age, while dependence on others increases [113]. The link between gender and the health locus of control is reflected by the higher scores obtained by men for the Internal Dimension, while the External Dimension dominates for women. This is associated with psychological differences formed at the prenatal stage, but primarily with the cultural construction of the sexes and stereotypes in operation [114]. In general, men are perceived as independent, and thus have a higher level of internal control. For women, there is often a perception of their dependence on others, or even helplessness, which may be expressed in a higher external locus of control [110]. It should be emphasized that gender differences in the location of health control are globally differentiated, with the smallest discrepancies observed in economically developed countries and the largest in cultures with low levels of gender egalitarianism [115].

The health locus of control may be related to above-normal body weight [116]. In the authors’ study, which included only obese individuals, it can be seen that they scored significantly lower on the Internal Dimension (22.7) and higher for the Chance Dimension (22.3), compared to the average values for Polish adults (25.5 and 20.6, respectively) [73]. Unfortunately, there has been no other research in Poland relating to the general population. However, studies on selected groups have shown that the average scores obtained for the Internal Dimension can range from 25.4 (people working in non-medical professions) [117] through 25.8 (students of health sciences) [118] to 26.2 (groups of healthcare professionals) [117] and for the Chance Dimension from 19.6 (people working in non-medical professions) [117] through 20.0 (students of health sciences) [118] to 20.2 (groups of healthcare professionals) [117]. Foreign research also supports the above findings. For healthy adults, the average values for the internal health locus of control are usually 24–26 [119,120,121,122], while for external control, with the Chance Dimension, are on average 17–22 [119,120,122]. The connection that exists between the health locus of control and body weight has been the subject of research for several decades. It has been suggested that the Internal Dimension is a predictor of successful obesity treatment; people with high internal control achieve more weight reduction and tend to maintain it longer, compared to the group with a dominant External Dimension. Meanwhile, overweight and obese individuals are more likely to have an external health locus of control [123,124]. Conversely, it has been observed that by knowing a patient’s attitude toward health control, obesity treatment programs can be adapted to their individual preferences, which will lead to greater satisfaction and more weight loss. Individuals with a strong Internal Dimension tend to get better results in individual programs, while with an external health locus of control, patients are more likely to prefer group-based programs [125]. Recent research has suggested that physicians should pay attention to the locus of health control in adolescent patients as well, as therapies oriented toward a greater internal dimension may significantly increase the success of eating disorder control [126].

In the relationship between obesity and health control, health behaviors play an important role [116,127]. Our research showed that internal health locus of control positively correlated with all health-promoting attitudes, increasing the possibility of adopting them, while external control, with a strong Chance Dimension, showed a negative correlation. The health locus of control influences the occurrence of obesity-related behaviors, from [128,129] the prenatal period through adulthood. Golding et al. [129] observed that a health locus of control, measured in pregnant women, may be important in the development of obesity in children. Children of mothers who scored high for the External Dimension during pregnancy were characterized by more fat mass in their teenage years (over 13 years). This may have been related to the behaviors represented by the women during pregnancy; women with an external health locus of control were more likely to smoke tobacco and abstain from breastfeeding. In addition, it was found that obtaining higher scores for the External Dimension in children preceded the development of obesity by at least 5 years [129].

A health locus of control is considered one of the personality traits. While the individual’s belief in the control of their own health has positive psychological and behavioral effects, the conviction that health depends on external factors can lead to feelings of loneliness and helplessness, and even to a higher incidence of mental disorders [130]. In our study, we observed that the external health locus of control, with the strong influence of the Chance Dimension, was associated with type D personality, whereas non-type D personality obtained significantly higher scores for the Internal Dimension. The present findings are consistent with other scientific reports. It is now thought that the relationship between distressed personality and improper eating behavior may be mediated by the location of health control [70,104]. Studies with chronic disease patients have shown that those with type D personality had poorer health treatment outcomes, due to low levels of self-control, projecting noncompliance with physician recommendations [131,132]. The connecting factor between the external health locus of control and type D personality would appear to be its stressful character. A 2018 study indicated that the implementation of an 8-week stress control program (known as an integrated relaxation technique program) had a significant effect on weight loss in obese patients. There were significant reductions in BMI, depression levels, and improvements in eating habits and physical activity. In addition, the group undergoing the stress control program was characterized by a decrease in scores for the subscale: Chance Dimension with a concomitant increase in the importance of Internal Dimension, resulting in a change in health attitudes [133]. The conducted research indicates that self-esteem of health control is not a constant trait; on the contrary, it is modifiable. Through health experiences, the future reactions of people are formed and, consequently, their attitudes toward disease and its treatment options [106,114].

## 5. Conclusions

Based on the results of the study, patients with distressed personality showed the lowest levels of overall health behaviors and were significantly different from those with intermediate and non-type D personalities in this regard. The authors indicated that type D personality was associated with poorer attitudes toward health, increasing the risk of improper health behaviors by more than five times. It was also observed that among the analyzed health behaviors, obese respondents with type D personality represented significantly the least effective preventive behaviors and mental attitudes. Moreover, among obese respondents with type D personality, there were significantly more respondents who believed that their health was a consequence of chance events, while there were the fewest respondents with a strong Internal Dimension. The internal health locus of control had the strongest positive correlation with all proper health behaviors.

Augmenting the diagnosis of patients with obesity with the identification of selected personality traits, taking into account the Type D personality, may improve their functioning and increase the chance of success of the applied weight reduction therapy.

## Figures and Tables

**Figure 1 ijerph-19-14650-f001:**
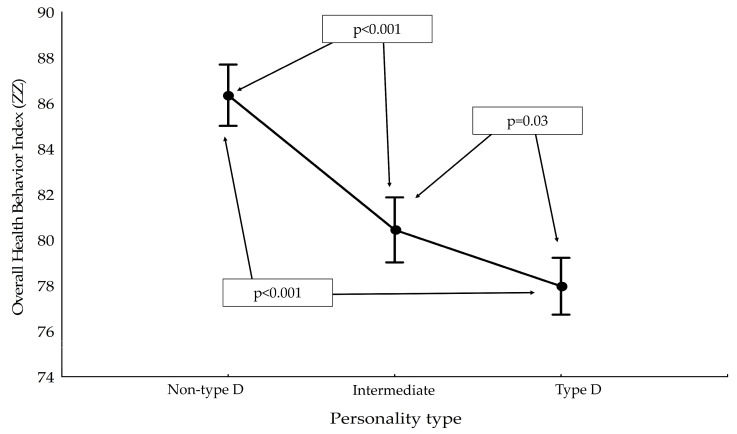
Results for the Overall Health Behavior Index (ZZ) by personality type.

**Figure 2 ijerph-19-14650-f002:**
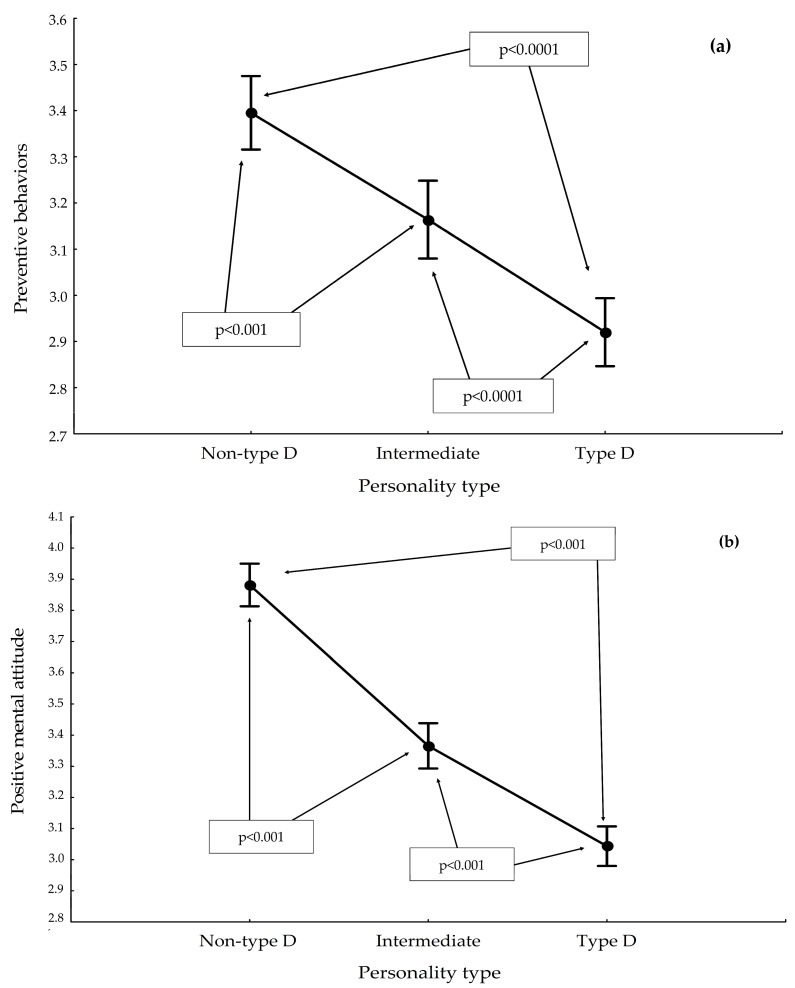
Results for selected subscales of IZZ, by personality type: (**a**) Preventive Behaviors subscale; (**b**) Positive Mental Attitude subscale.

**Figure 3 ijerph-19-14650-f003:**
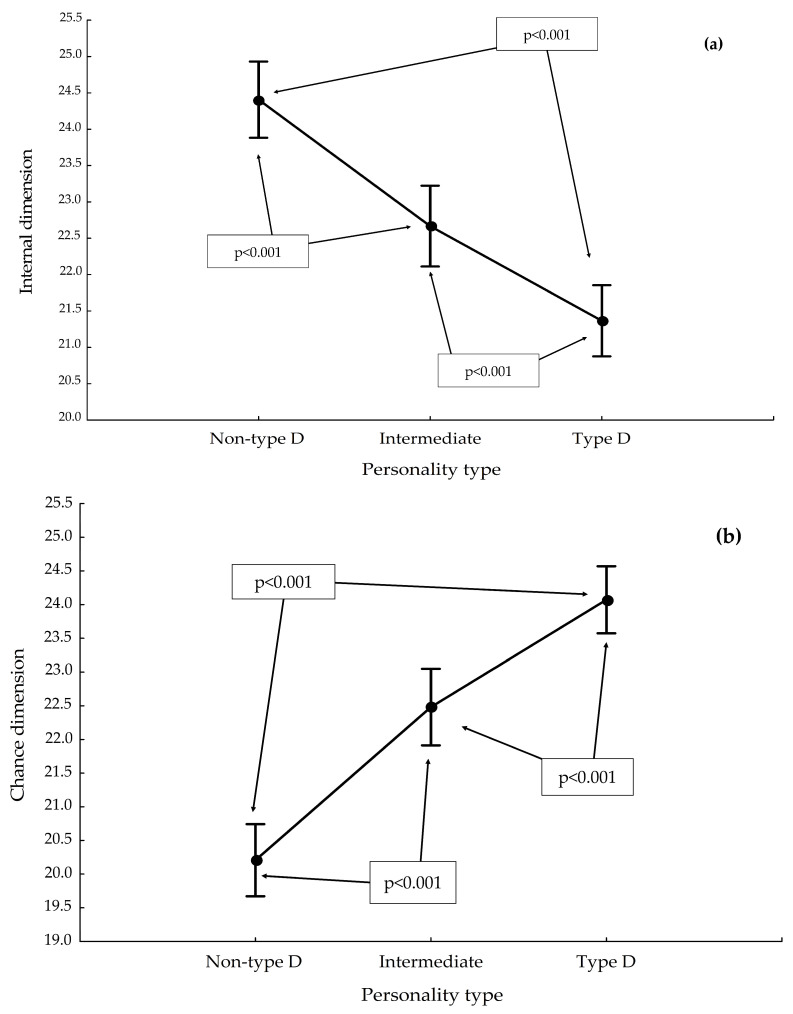
Results for selected subscales of MHLC-A, by personality type: (**a**) Internal Dimension; (**b**) Chance Dimension.

**Table 1 ijerph-19-14650-t001:** Norms for The Inventory of Health Behaviors (IZZ) to convert the Overall Health Behavior Index (ZZ) to sten scores (based on: [73]).

Male	Sten Scores	Female
ZZ (Score)		ZZ (Score)
24–50	1	24–53
51–58	2	54–62
59–65	3	63–70
66–71	4	71–77
72–78	5	78–84
79–86	6	85–91
87–93	7	92–98
94–101	8	99–104
102–108	9	105–111
109–120	10	112–120

ZZ—Overall Health Behavior Index.

**Table 2 ijerph-19-14650-t002:** The interpretation of the results for Type D Scale (DS-14) (based on: [75,76]).

Personality Type	Total Scores for the Subscale	
	Negative Affectivity	Social Inhibition
Type D personality	≥10	≥10
Intermediate personality	≥10	<10
	<10	≥10
Non-type D personality	<10	<10

**Table 3 ijerph-19-14650-t003:** Scores obtained for the ZZ and individual subscales of the IZZ in our study compared with the values for the general adult population in Poland [73].

IZZ	Value from This Study X ± SD	95% CI of the Mean (X)	Normative Value X ± SD	t	p^t^
Overall Health Behavior Index (ZZ)	81.4 ± 9.0	80.6–82.3	81.82 ± 14.16	−0.91	>0.05
Proper Eating Habits	3.5 ± 0.6	3.4–3.5	3.22 ± 0.76	10.84	<0.0001
Preventive Behaviors	3.1 ± 0.5	3.0–3.1	3.42 ± 0.78	−10.94	<0.0001
Positive Mental Attitude	3.4 ± 0.5	3.3–3.4	3.52 ± 0.66	−4.16	<0.0001
Health Practices	3.4 ± 0.4	3.4–3.5	3.32 ± 0.85	6.97	<0.0001

CI—confidence interval; p^t^—Student’s t-test for a single sample; t—t-value; X ± SD—mean ± standard deviation.

**Table 4 ijerph-19-14650-t004:** Factors that influence the undertaking of health-promoting behaviors; results of logistic regression (backward selection).

Dependent Variable—Health Behavior	Factor—Predictor	Predictor Characteristics	OR (95% CI)
The undertaking of health-promoting behaviors	Age (years)	<36	1
		31–40	0.64 (0.25–1.64)
		41–50	0.19 (0.08–0.48)
		51–60	0.12 (0.05–0.29)
		61–70	0.09 (0.03–0.23)
		>71	0.04 (0.01–0.13)
	Personality type	Non-type D	1
		Intermediate	0.32 (0.18–0.56)
		Type D	0.18 (0.10–0.32)

CI—confidence interval; OR—odds ratio.

**Table 5 ijerph-19-14650-t005:** Average scores for each subscale of the MHLC-A considering the socio-demographic data of the study group.

	Internal Dimension Me (Q1–Q3)	Powerful Others Dimension Me (Q1–Q3)	Chance Dimension Me (Q1–Q3)
Gender			
Female	22.0 (21.0–24.0)	23.0 (22.0–25.0)	23.0 (19.0–24.0)
Male	22.0 (21.0–25.0)	22.0 (20.0–24.0)	23.0 (21.0–25.0)
p^M-W^	0.9	<0.0001	0.1
Age (years)			
≤30	26.0 (23.0–28.0)	22.0 (21.0–24.0)	23.0 (19.0–24.0)
31–40	24.0 (22.0–27.0)	22.0 (21.0–24.0)	22.0 (19.0–24.0)
41–50	23.0 (22.0–25.0)	23.0 (21.0–24.0)	22.0 (18.0–24.0)
51–60	22.0 (20.0–23.0)	23.0 (22.0–25.0)	23.0 (21.0–25.0)
61–70	21.0 (20.0–22.0)	23.5 (22.0–25.0)	24.0 (22.0–26.0)
>70	20.0 (18.0–22.0)	22.0 (20.0–26.0)	23.0 (22.0–26.0)
Wartość p^K-W^	0.0001 ^1^	0.07	0.003 ^2^
Place of residence			
Village	23.0 (21.0–25.0)	22.0 (20.0–24.0)	23.0 (19.0–25.0)
City, with populations <50,000	23.0 (17.0–25.0)	22.0 (21.0–23.0)	24.0 (22.0–25.0)
City, with populations 50,000–100,000	22.0 (19.5–23.0)	22.0 (21.0–24.0)	23.0 (20.0–24.0)
City, with populations 100,000–250,000	22.0 (20.0–24.0)	23.0 (22.0–25.0)	23.0 (20.0–25.0)
City, with populations >250,000	23.0 (21.0–25.0)	23.0 (21.0–25.0)	22.0 (20.0–24.0)
Wartość p^K-W^	0.051	0.07	0.5
Educational level			
Primary education	22.0 (20.0–23.0)	23.0 (21.0–25.0)	23.0 (21.0–25.0)
Professional education	22.0 (21.0–24.0)	23.0 (21.0–24.0)	23.0 (20.0–25.0)
Secondary education	22.0 (21.0–24.0)	23.0 (21.0–25.0)	23.0 (21.0–25.0)
Higher education	23.0 (21.0–25.0)	23.0 (21.0–25.0)	22.0 (19.0–24.0)
Wartość p^K-W^	0.4	1.0	0.3
Classes of obesity			
Obese class I (BMI: 30.0–34.9 kg/m^2^)	22.0 (21.0–24.0)	23.0 (21.0–25.0)	22.0 (20.0–25.0)
Obese class II (BMI: 35.0–39.9 kg/m^2^)	22.0 (21.0–25.0)	23.0 (21.0–24.0)	22.0 (20.0–25.0)
Obese class III (BMI: ≥40.0 kg/m^2^)	22.0 (21.0–24.0)	23.0 (21.0–24.0)	23.0 (20.0–25.0)
Wartość p^K-W^	0.4	0.9	0.6

BMI—Body mass index; Me—median; p^K-W^—Kruskal–Wallis test with post hoc analysis; p^M-W^—Mann–Whitney U test; Q1—first quartile; Q3—third quartile. ^1^—post hoc analysis: ≤31 vs. 41–50, 51–60, 61–70, >71; 31–40 vs. 51–60, 61–70, >70; 41–50 vs. 51–60, 61–70, >70; 51–60 vs. >70. ^2^—post hoc analysis: 31–40 vs. 61–70, >70; 41–50 vs. 61–70.

## Data Availability

The data presented in this study are available on request from the corresponding author.

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
