# Peer review of "Type D Personality and Health Behaviors in People Living with Obesity"

_ijerph, 2022, doi:10.3390/ijerph192214650_

Round 1

Reviewer 1 Report

This paper investigates the role of type D personality on health behavior in Polish obese patients in comparison with intermediate and non-type D personality using a number of scales in response to the completion of 3 standardized questionnaires. The findings are that type D personality in obese respondents is associated with poorer attitudes towards health, including the belief that their health status is a consequence of chance events. In obese patients, this study suggests that knowing their personality type may improve the appropriateness of the program chosen for weight loss and increase the chance of success of weight loss therapy.

The paper is well-written, reporting a well-conducted study. The points made are clear, insightful and well-argued. However, there is much new research on obesity and the authors have to ensure their references are no older than 2018 unless the reference cited is the seminal work in the area.  As such, the majority of the references need to be updated to provide the most current research available and the arguments presented must correspond to this updated review of the literature. The Discussion is referenced to a higher standard than the Introduction, but still requires work in updating references.

Line by line suggested edits

37-39 Need at least one reference for these claims. 

41-47 Rather than cite two references at the end of these three sentences, indicate the reference for each statement. Given that this paper presents cutting edge research, quoted statistics regarding obesity need to be up to date. The references for these statistics are too old.

48-59 There are many claims made in these six sentences; yet, rather than provide a citation for each claim, three citations are offered at the end as representing the references for these claims. The references are too old, especially the one from 2009.

59-65 Citation 5 is to research published in 2014. There is has been a significant amount of research done on the five-factor personality model and obesity since that time. The authors must update their review of this literature and, based on this updated review, provide the reason why a focus on type D personality is more appropriate than making use of the five-factor personality model when considering health behavior in obese patients.

67 Given that this paper is based on an understanding of type D personality, please begin this discussion of type D personality with a history of the development of this term in personality research. 

67-71 References 7 and 8 are not only too old to be relevant to this current discussion, they are also not directly related to obesity, rather, they concern diabetic foot syndrome and chronic pain. Make sure that the references to support the claims with respect to health behavior are directly relevant to obesity.

71-79 There are so many papers that have been published in peer-reviewed journals since 2018 on type D personality and obesity that there is no reason for the authors to look to either outdated studies or ones related to breast cancer (as is the only study referenced that is up-to-date).

80-88 There are many claims in these four sentences and only one reference from 2017 provided to support these claims. Make sure that each claim is referenced and that the reference is no older than 2018. There is a great deal of research that has been done since 2018 regarding the hippocampus and obesity, for example.

91-98 Is the claim that Gochman and Parkerson provided the seminal research in these areas? If not, why are the authors quoting their outdated work? If the authors are referring to this work because it is the original work in this area, they need to tell this to the reader. Otherwise, it is important to update the references on the complexity of health behavior and how it is viewed.

98-100 A book published in 2008 cannot be cited as providing a “current” basic definition of health behavior. Find a more recent reference published in a peer-reviewed journal.

99 Change “behavior, states” to “behavior states”.

101 Change “It has been difficult to define health behaviors and has influenced differences” to “A difficulty in defining health behaviors has influenced differences”.

102 Were Kasl and Cobb the first to define 3 categories of health behaviors in 1966? if so, reference their original work, not a secondary source from 2008.

108-110 If there have been other ways of classifying health behaviors the citations need to be to the original papers making these classifications, not to an outdated text from 2008. It is also best that the authors indicate what ways of classifying behaviors are considered most relevant today by citing research since 2018.

121-126 This ending paragraph needs to explain why, of all the ways of classifying health behaviors, the authors chose to examine Type D personality. What was their criteria in making this selection?

143 How was consent obtained from those who participated? 

145 What was the incentive for patients to participate in the study?

188 Please provide the citation to the original research by Wallston, Wallston, and Devillis.

211 Who developed the DS-14? Please provide a citation to the original research.

228 Citation 23 is not to a reference regarding the Shapiro-Wilk test. Please provide an appropriate reference to this test.

229 Please provide a reference for Levene’s test and for Tukey’s correction.

232 Please provide a reference for Spearman’s nonparametric correlation test.

236 Please provide a reference for STATISTICA 13.0 PL.

273-275 References are needed for both the Health Behavior Inventory and the Overall Health Behavior Index. If these are merely titles, say so.

287-288 The resolution for Figure 1 is too low. Please provide a higher resolution picture of Figure 1.

304-305 The resolution for Figure 2 (a) is too low. Please provide a higher resolution picture of Figure 2 (a).

305-306 The resolution for Figure 2 (b) is too low. Please provide a higher resolution picture of Figure 3.

334 Please provide a reference for MHLC-A.

364-365 The resolution for Figure 3 (a) is too low. Please provide a higher resolution picture of Figure 3 (a).

365-366 The resolution for Figure 3 (b) is too low. Please provide a higher resolution picture of Figure 3 (b).

392 Please provide a reference no earlier than 2018.

405 Please provide a reference no earlier than 2018.

425 Reference 29 is out of date. Please provided an updated reference.

427-433 Reference 20 is out of date. Please provided an updated reference.

434-435 Please provide at least one reference for these claims.

486-514 Except for citation 45, which is stated as being from 2017, the rest of the citations are presented as current research by the text. All of these citations are to research that is out of date and the research needs to be updated.

515-526 There are a number of claims made in this paragraph and only one citation to an out of date paper. Please provide citations to current references for all claims.

535-538 These citations are to out of date references. Please provided current references. 

543-552 There are a number of claims made in these four sentences and only the first reference is to current research. Please provide current references for each claim after the sentence where the claim is made, not as a group at the end of the paragraph.

553 Need a reference for this claim.

553-570 All references cited in this paragraph, except for 57, are too old and need to be updated.

571-572 Need a reference for this claim. 

576-579 Unless more up to date data can be found on this topic, please eliminate this point. The research was published in 2008 and is otherwise irrelevant to the discussion as it concerns knowledge of illness during childhood: making distinctions between cancer and colds.

588 Please provide a reference for this claim.

588-611 Except for references 62 and 65, all other citations in this paragraph are to references that are out of date. Please update the references.

Please provide the DOI for references in the following lines (for those references that are to be retained once the references are updated): 649, 658, 677, 679, 687, 691, 700, 704, 706, 708, 747, 755, 765, 767, 773, 779, and 783.

Author Response

Dear Reviewer, 

we would like to thank you for taking your time to review, and we are sure that improving this paper based on your comments will significantly increase its merit. In response to your comments, we would like to point out that changes have been made to the text as indicated. In the attached file we refer to the individual comments made in the review. 

Reviewer 2 Report

Reviewer comments

In this work, Buczkowska et al. investigated the association between type D personality and health behaviors in people living with obesity. Kindly, find below my comments for your response.

Title: It is not ideal to use the condition of participants to describe them. For example, hypertensives, diabetics and obese patients. Consequently, I want to suggest that the authors tweak the title to “Type D personality and health behaviors in people living with obesity”

Introduction

Line 51-52: Kindly, provide a reference

Line 80-85: Kindly, provide reference

The authors should provide references to support the statements presented.

Materials and Method

Line 130: Please, on what grounds was the sample size set at 400?

My major problem with this work is the lack of Ethical approval. How do the authors guarantee the safety of participant information? How was the participant information stored?

Results

Where in the work was the logarithmic transformation carried out? I can’t see that in any of the results section yet the authors have presented in the “Statistical analysis section” that “A logarithmic transformation was used to normalize the values of selected parameters (if necessary).”

Discussion

The authors should discuss the results of the study. I find the first paragraph of the Discussion as almost an Introduction. I suggest that the authors rather discuss the results of the work.

Conclusion

The authors should kindly present the conclusion as full paragraph and not in bullets.

Author Response

Dear Reviewer, 

we would like to thank you for taking the time to review it, and we are sure that improving this paper based on your comments will significantly increase its merit. In response to your comments, we would like to point out that changes have been made to the text as indicated. In the attached file we refer to the individual comments made in the review. 

Round 2

Reviewer 1 Report

The authors of this excellent study are thanked for making the corrections suggested by this reviewer. Each that has been made has improved the relevance of the paper considerably, primarily by updating the references.

Regarding those references that were not updated by the authors because no more recent research has been found on the topic, if the research is older than five years, it is best that the authors let readers know that this is the most recent research available. After five years, conditions change; when research hasn’t been undertaken with respect to a specific topic in the last five years, it becomes questionable consequently.

Now that the Introduction and Discussion are written to correspond to the most recent research on obesity, there are a few additional suggested edits that will make the paper then ready for publication. These mainly relate to changes in wording and statements that have been made in the Introduction that require the support of references that are missing. Furthermore, paragraphs in the Materials and Methods section need rearranging and an additional subheading. Please also note that all numbering of the section subheadings should follow MDPI style. In other words, each number must end in a period, e.g., 2.1 must be changed to 2.1., to correspond to the style; as well, be careful to ensure that the references are standardized in their layout.

I enjoyed re-reviewing this paper and look forward to its imminent publication.

Line by line suggested edits

22 Change “worst” to “least effective”.

43- It is more informative to change “corresponding to about USD PPP (purchasing power parity) 209 per capita per year” to the full information from reference 6, “corresponding to approximately  USD PPP (purchasing power parity) 311 billion per year (or USD PPP 209 per capita per year)”.

44-46 “The most important consequences of obesity are chronic diseases: diabetes, cardiovascular disease, and cancer.”—please provide a reference for this statement, it may the reference cited in the Discussion as 69. 

46-47 “By 2050, obesity will be responsible for 70% of all diabetes treatment costs, 23% of cardiovascular disease treatment costs and 9% of cancer treatment costs.”—please provide a reference for this statement. 

63 Change “despite the same skills” to “despite possessing the same skills”

121 Change “dysfunction are exacerbated” to  “dysfunctional, which are exacerbated”.

150-151 “Health behavior is strongly linked to our lifestyle, which is derived from individual predispositions and social and cultural conditions.”—please provide a reference for this statement, the reference may be either what is later cited as 48 or 55 in the Discussion.

151-152 “Health habits are formed during early life, influencing later behavior and, consequently, the health of the adult.”—please provide a reference for this statement. The reference is likely the one cited as 68 in the Discussion.

152-154 “The health behaviors undertaken depend on age, gender, education, marital status, family, and financial situation and occupation.”—please provide a reference for this statement. Regarding gender, the reference is likely 58, as cited in the Discussion. Age would likely be either 87 or 88, as cited in the Discussion.

154-155 “Personality traits also play an important role in determining health behavior”—please provide a reference for this statement, the reference is likely either 75 or 76—as cited in the Discussion.

155-157 “It has been observed that risky behaviors are associated with low-stress tolerance, emotional immaturity, difficulty expressing feelings, low self-esteem, feelings of loneliness, and high levels of anxiety.” —please provide a reference for this statement. Some might be 78, 82, 83 and/or 86, as mentioned in the Discussion.

160 Change “Considering the association of type D personality with worse attitudes” to “Given the association of type D personality with the least effective attitudes”. 

165 Change “common those” to “common among those”.

167 Change “in a group of obese patients” to “in a group of obese patients in Poland, where more than half of all adults have above-normal body weight”.

172 Change “survey” to “survey of the study”.

176-178 Please move “The study group was adult patients with BMI ≥30 kg/m2, who were hospitalized in selected hospitals in Silesia Voivodeship, Poland.” to be the first sentence of the first paragraph of 2.1. 

177-178 Create a new subheading 2.2. Ethics approval. Make “The study design was approved by the directors of the institutions in which the study was conducted.” the first sentence of this new subheading.

179-182 Move this paragraph up to be the last paragraph of the previous subheading, 2.1. Study group. In other words, this subsection should read as follows:

The study group was adult patients with BMI ≥30 kg/m2, who were hospitalized in 176 selected hospitals in Silesia Voivodeship, Poland.

The survey of the study was conducted between May 2018 and December 2019. A total of 443 correctly completed questionnaires were analyzed. Based on the unknown fraction of type D personality in the population of hospital patients, and assuming an effect size (fraction) of at least 0.1 [57] , the minimum sample size was set at 440.

Exclusion criteria were a condition that made it impossible to complete the questionnaires, including severe disease accompanied by fever or a postoperative condition, dependence, and symptoms of impaired cognitive functioning (autopsychic and allopsychic orientation disorders identified by screening).

183-191 Please reorganize this information under the new subheading 2.2. Ethics approval as follows:

The study design was approved by the directors of the institutions in which the study was conducted. Bioethics Committee approval was not required, due to the questionnaire type of this study (opinion dated 23/05/2018 No.: KNW/0022/KB/106/18). According to Polish law, this study was not a medical experiment, so it did not require the consent of the Bioethics Committee (Act of December 5, 1996, on the professions of physician and dentist (i.e. Journal of Laws 2019, item 537). Nevertheless, all research standards were observed in the study. It complies with the provisions of the Declaration of Helsinki. Patient contribution to the study was voluntary. The authors used no patient identification or confidential data from medical records. The headline of each survey contained a note to the patient that the survey was anonymous and the results would be used for research purposes.

192 Change “2.2” to “2.3.

196 Change "2.1.1" to "2.3.1."

200 Change "2.1.2" to "2.3.2." 

235 Change "2.1.3" to "2.3.3."

258 Change "2.1.4" to "2.3.4."

271 Change "2.2" to "2.4."

484 Change “giant obese” to “morbidly obese, defined as those with  BMI ≥ 40 kg/m 2 [72]”. Making this change will require a new reference defining morbidly as a technical term:

Moon, T.S.; Fox, P.E.; Somasundaram, A.; Minhajuddin, A.; Gonzales, M.X.; Pak, T.J.; Ogunnaike, B. The influence of morbid obesity on difficult intubation and difficult mask ventilation. J. Anesth. 201933, 96-102. https://doi.org/10.1007/s00540-018-2592-7

486 Change “those with giant obesity” to “the morbidly obese”. 

499-500 Change “the worst mental attitude (3.0 sten) (Positive Mental Attitude subscale) and the worst preventive” to “the least effective mental attitude (3.0 sten) (Positive Mental Attitude subscale) and the least effective preventive”.

552 Change “The best health behaviors were represented by those younger than 36, and the worst by patients” to “The most effective health behaviors were represented by those younger than 36, and the least effective by patients”

558-559 Change “there was observed that the recommended levels of physical activity were most rarely achieved in the group over 65 years old then 45-64 and the most active were those in the” to “it was observed that the recommended levels of physical activity were most rarely achieved in the group over 65 years old and then the group 45-64 years old. The most active were those in the”.

675 Change “right and wrong health experiences” to “health experiences”. Health experiences are neither moral issues nor statements of truth; therefore, they cannot be judged as right or wrong.

685 Change “the worst preventive behaviors and mental attitudes” to “the least effective preventive behaviors and mental attitudes”.

686 Change “re-spondents” to “respondents”.

References

Please line up the following references with the left margin:

1, 5, 8, 10, 17, 19, 34, 36, 44, 49, 50- 54, 56, 58, 60, 61, 68-74, 77-89, 91-93, 96-100, 102-104, 108-109, 111, 113, 116, 118, 120.

References missing the DOI number: 50, 56, 57, 73, 74, 109, 120. If a reference doesn’t have a DOI number, the link where is it available and the date it was accessed need to be mentioned.

This reference is missing a number and likely is not cited in the paper as a result, 

Gore, J.S.; Griffin, D.P.; McNierney, D. Does Internal or External Locus of Control Have a Stronger Link to Mental and Physical Health? Psychol Stud (Mysore) 201661, 181–196, doi:10.1007/s12646-016-0361-y.

Author Response

Dear Reviewer,
thank you very much for taking the time to review it again. All the suggestions posted are very valuable. We are very grateful for such precise guidelines relating to the text, which made it easy to make personal changes. We have responded to all suggestions below.

Reviewer 2 Report

Thank you for undertaking the revisions. My major concern relates to the authors response to why no Ethical approval was sought for this work. Yes, I understand the authors purely administered a questionnaire to the participants. Good standards of ethical practice in research still requires that Ethical approval is sought from the participants. I asked about how the participants information was kept confidentially but the authors did not respond to that. The authors made this statement "Bioethics Committee approval was not required, due to the questionnaire type of this study (opinion dated 23/05/2018 No.: KNW/0022/KB/106/18)" in the manuscript. I would like to know what the "opinion" meant. Is it opinion from the participants or from the Bioethics committee?

Author Response

Dear Reviewer,
of course, each patient gave informed consent to participate in this study. Opinion dated 23/05/2018 No.: KNW/0022/KB/106/18 was provided by the Bioethics Committee. 
The questionnaires completed by the participants are kept at the place of employment of the first author of the publication (Department of Toxicology and Health Protection - a locked cabinet) and this is the only person who has access to them. The database that was created based on the completed surveys is stored only on company equipment, which is password protected.